# Control of Linear Astigmatism Aberration in a Perturbed Axially Symmetric Optical System and Tolerancing

**José Sasián**

Wyant College of Optical Sciences, University of Arizona, 1630 E University Boulevard, Tucson, AZ 85721, USA; JOSE.SASIAN@OPTICS.ARIZONA.EDU

**Abstract:** Linear astigmatism aberration is undesirable because it rapidly degrades image quality. We discuss some techniques to control and mitigate this aberration, and provide a comparison of lens systems that have been desensitized for linear astigmatism aberration.

**Keywords:** linear astigmatism; lens tolerancing; lens desensitizing





## 1. Introduction

Axially symmetric lens systems are often required to be aplanatic to provide sharp imaging at least in a central portion of the system's field of view. The current meaning of the term aplanatic is freedom from spherical aberration and coma aberration. However, the meaning can be defined as absence of uniform and linear field-dependent aberrations. Aplanatism is a very desirable property because it provides sharp imaging and ease of lens alignment. Because of lens fabrication and assembly errors, there are not in practice perfect axially symmetric systems but perturbed optical systems [1–3].

When the axial symmetry is lost the first aberrations that can be present, in addition to the aberrations of axially symmetric systems, are uniform astigmatism, uniform coma, linear astigmatism, field tilt, and smile and keystone distortion [4]. Then, under perturbation and to keep aplanatism one must correct for uniform astigmatism, uniform coma, and linear astigmatism aberration. Because uniform astigmatism depends on the square of the surfaces tilt with respect to the nominal optical axis, and given small system perturbations, then uniform astigmatism turns out to be negligible. If there is any significant uniform astigmatism it is likely due to surface figure error or lens strain. Thus to keep the aplanatism of a perturbed aplanatic system one must correct or mitigate for uniform coma and linear astigmatism.

Uniform coma, sometimes called central coma [5], is often a problem in high numerical aperture systems and can be corrected by decentering a lens element. This has been done in assembling microscope objectives and in some early projection lenses for micro lithography. Another way to correct for uniform coma is reprocessing a system surface at the stop aperture location to introduce the opposite amount of uniform coma and rely on coma cancellation. Still another way to mitigate for uniform coma is to disassemble and reassemble the lens system several times until uniform coma is found to be reduced. In this case one relies on the statistical nature of the lens aberration residuals as a function of lens assembly.

On the other hand linear astigmatism [6] is a problem in wide angle lenses. This paper discusses several methods to mitigate linear astigmatism aberration in a system that is nominally axially symmetric but that due to fabrication and assembly errors suffers from this aberration.

There has been interest in understanding the behavior of perturbed lens systems in relation to tolerancing a nominally axially symmetric system [7–15]. Our study focuses on the control of linear astigmatism aberration. Other techniques for aberration mitigation

in projection lenses not discussed in this paper are using off-axis illumination and image processing [16].

## 2. Materials and Methods

A first step is to determine lens sensitivity to uniform astigmatism $W_{UA}$, uniform coma $W_{UC}$, linear astigmatism $W_{LA}$, and field tilt $W_{FT}$. These aberrations are calculated as a single surface in the lens is tilted a small amount such as 0.1°. Equations (1)–(4) are used to calculate the aberrations using an on-axis real ray, and two real rays, +1 and −1, at full aperture in the plane of symmetry of the perturbed system. Then the lens sensitivity to the aberrations is calculated as the RSS (root sum square) of the aberrations contributed by each tilted surface.

$$W_{UC} = \frac{OPL(+1) - OPL(-1)}{2} \tag{1}$$

$$W_{UA} = 8\frac{(SAGT(0) - SAGS(0))}{F^2} \tag{2}$$

$$W_{LA} = 8\frac{(SAGT(+1) - SAGS(+1)) - (SAGT(-1) - SAGS(-1))}{2F^2} \tag{3}$$

$$W_{FT} = 8\frac{SAGS(+1) - SAGS(-1)}{2F^2} \tag{4}$$

In Equations (1)–(4), OPL stands for the optical path length, SAGT stands for the sag of the tangential field curve, SAGS stands for the sag of the sagittal field curve, and F stands for the f-number of the system so that the aberration is given as a wavefront error.

Keeping in mind that tilting a single surface reduces the system symmetry to plane symmetry, the calculation of uniform coma requires the OPL of two rays in the plane of symmetry. Since lens design software can provide directly the sag of the astigmatic field curves in the plane of symmetry, then the calculation of uniform astigmatism, linear astigmatism, and field tilt is simplified. A discussion of these aberrations is found in reference [4].

The choice of 0.1° surface tilt is because this amount of surface tilt often separates what is easy to manufacture from what is difficult to manufacture. The sensitivity analysis could be done with a larger surface tilt and the results would be similar because uniform coma, linear astigmatism, and field tilt are proportional to surface tilt.

Depending on how the data is processed Zernike polynomials or other type of polynomials can be used. However, here for clarity we chose a simple aberration representation.

The analyses discussed below were done with Zemax and OpticStudio lens design software.

### 2.1. Lens Desensitizing

A first method to control linear astigmatism is to desensitize a lens. Consider a Cooke triplet lens as shown at left in Figure 1 with a field of view of ±24°, focal length of 50 mm and an optical speed of F/5. The sensitivities of this lens are shown in Table 1. As discussed above there is no significant sensitivity to uniform astigmatism, and linear astigmatism has the largest sensitivity.

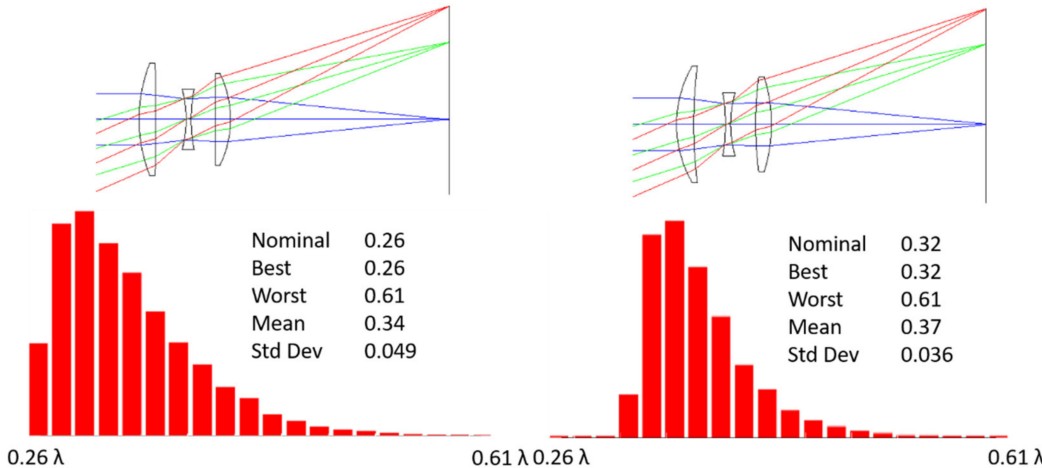

**Figure 1.** **Left**, Cooke triplet lens; **right**, desensitized Cooke triplet lens. Histograms for 1000 Monte Carlo trials of rms (root mean square) wavefront error across the field of view.

**Table 1.** Cooke triplet lens surface sensitivities in waves.

| Surface | \multicolumn Cooke Triplet Lens Surface Sensitivities in Waves | | | |
|---|---|---|---|---|
| | $W_{UA}$ | $W_{UC}$ | $W_{LA}$ | $W_{FT}$ |
| 1 | 0.00 | −0.12 | 0.03 | 1.04 |
| 2 | 0.00 | −0.14 | 2.71 | −0.27 |
| 3 | 0.00 | −0.36 | −2.15 | −0.57 |
| STOP | 0.00 | 0.22 | −1.57 | −0.39 |
| 5 | 0.00 | −0.19 | 0.52 | −0.71 |
| 6 | −0.01 | 0.47 | 1.15 | 0.80 |
| RSS | 0.01 | 0.68 | 4.00 | 1.67 |

To desensitize the Cooke triplet lens to linear astigmatism it was re-optimized including in the error function the lens sensitivity to this aberration by means of a macro language routine that provided the aberration sensitivity to the lens design program optimizer. The resulting lens cross section is shown on Figure 1 at right and the lens sensitivities are given in Table 2. As can be seen the sensitivity to aberrations has been reduced.

**Table 2.** Desensitized triplet lens surface sensitivities in waves.

| Surface | \multicolumn Desensitized Cooke Triplet Lens Surface Sensitivities in Waves | | | |
|---|---|---|---|---|
| | $W_{UA}$ | $W_{UC}$ | $W_{LA}$ | $W_{FT}$ |
| 1 | 0.00 | −0.17 | −0.06 | 1.09 |
| 2 | 0.00 | 0.10 | 1.35 | −0.59 |
| 3 | 0.00 | −0.25 | −1.89 | −0.41 |
| STOP | 0.00 | 0.20 | −1.02 | −0.40 |
| 5 | 0.00 | −0.13 | 0.90 | −0.40 |
| 6 | 0.00 | 0.26 | 1.24 | 0.65 |
| RSS | 0.01 | 0.48 | 2.96 | 1.57 |

Figure 1 also shows the histograms of 1000 Monte Carlo trials when only the surfaces were allowed to tilt 0.1° during a tolerance analysis. For the Cooke triplet lens at left the nominal performance is 0.26 waves rms, the best trial also had a performance of 0.26 waves rms, the worst trial 0.61 waves rms, the mean trial 0.34 waves rms, and the standard deviation was 0.049 waves rms. In comparison, the desensitized Cooke triplet lens had a nominal performance of 0.32 waves rms, the best trial also had a performance of 0.32 waves rms, the worst trial 0.61 waves rms, the mean trial 0.37 waves rms, and the standard deviation was 0.036 waves rms. Previously the desensitized form of a Cooke

triplet in Figure 1 at right has been reported [17]; here we note that the main problem is linear astigmatism.

### 2.2. Lens Design form Change

Figure 2 shows a triplet lens with the same field of view, focal length, and focal ratio like the lens in Figure 1, but with a different lens form because of compactness. The lens in Figure 2 at left has an improved nominal performance of 0.17 waves rms across the field of view. A Monte Carlo tolerancing analysis with only surface tilts of 0.1° resulted in a best trial of 0.19 waves rms, a worst trial of 0.87 waves rms, a mean trial of 0.38 waves rms, and a standard deviation of 0.13 waves rms. Although this triplet lens has a significant improved nominal performance, under mass production, it may not be preferable to the lens in Figure 1 because of the larger standard deviation of 0.13 waves rms. By desensitizing the lens for linear astigmatism as discussed above, the lens in Figure 2 at right results. A Monte Carlo analysis of this desensitized lens results in a nominal performance of 0.32 waves rms, a best trial of 0.32 waves rms, a worst trial of 0.72 waves rms, a mean trial of 0.40 waves rms, and a standard deviation of 0.06 waves rms. Figure 2 also provides histograms for 1000 Monte Carlo trials versus the rms wavefront error across the field of view. This compact lens shows that a better nominal performance may not be preferable because the as-built lens would have a much larger standard deviation albeit similar mean performance. However, if only a few lenses are made and compensation is allowed, then this compact lens might be preferable because the nominal performance of 0.17 waves rms could be obtained.

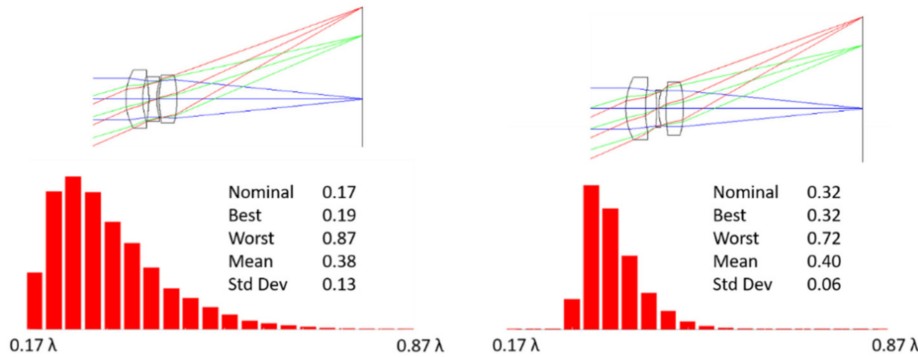

**Figure 2. Left**, triplet lens; **right**, desensitized triplet lens. Histograms for 1000 Monte Carlo trials of rms wavefront error across the field of view.

### 2.3. Retrofocus Lens

A third example of desensitizing a lens is the retrofocus lens in example 1 of US Patent 2,824,495 to G. Klemt [18] with a field of view of ±31°, focal length of 50 mm, and an optical speed of F/4. This lens is shown in Figure 3 at left where the lens was reoptimized with modern lens design software. At right in Figure 3 is the lens desensitized for linear astigmatism as discussed above. Monte Carlo analysis of both lenses using only surface tilts of 0.1° as lens errors, were performed as also illustrated at bottom in Figure 3 with histograms of 1000 trials. This retrofocus lens case highlights better the results of desensitizing a lens for a given effect, in this case linear astigmatism. The desensitized lens has a nominal performance degraded almost by a factor of two. However, the trials mean for both lenses is about the same but the desensitized lens has about half the standard deviation. In addition the desensitized lens does not produce either very good performing trials, or very bad performing trials. Thus for mass production and quality control a desensitized lens might be preferable to one optimized solely for image quality.

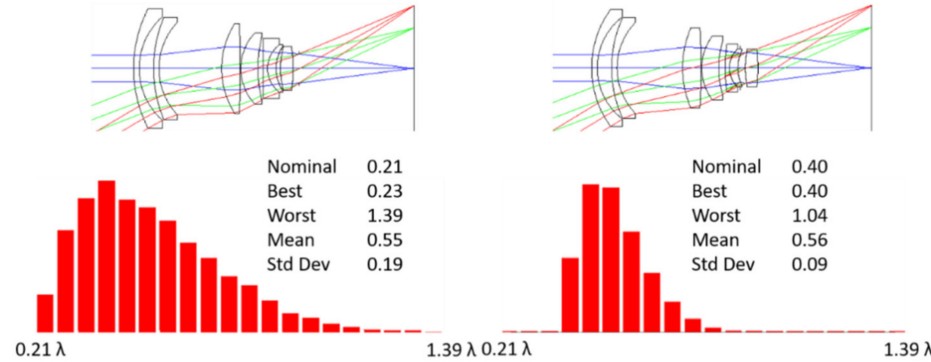

**Figure 3. Left**, retrofocus lens; **right**, desensitized retrofocus lens. Histograms for 1000 Monte Carlo trials of rms wavefront error across the field of view.

Tables 3 and 4 provide the sensitivities of lenses in Figure 3. In the desensitizing process the sensitivity to linear astigmatism was reduced from 9.91 waves to 5.63 waves. In addition, the sensitivities to uniform coma and field tilt were also reduced.

**Table 3.** Retrofocus lens. Surface sensitivities in waves.

| Retrofocus Lens Surface Sensitivities in Waves | | | | |
|---|---|---|---|---|
| Surface | $W_{UA}$ | $W_{UC}$ | $W_{LA}$ | $W_{FT}$ |
| 1 | 0.00 | −0.05 | −0.05 | −0.05 |
| 2 | 0.00 | 0.19 | 1.11 | 0.48 |
| 3 | 0.00 | −0.08 | 0.82 | 0.15 |
| 4 | 0.00 | 0.38 | 2.46 | 1.48 |
| 5 | −0.01 | −1.43 | −3.03 | −1.85 |
| 6 | −0.01 | 0.58 | 2.01 | 0.95 |
| 7 | 0.00 | −0.71 | −0.53 | 0.64 |
| 8 | 0.00 | 0.38 | 1.67 | −0.28 |
| 9 | 0.00 | −0.90 | −5.97 | −1.58 |
| 10 | 0.00 | −0.15 | −0.27 | −0.04 |
| 11 | 0.00 | 1.31 | 0.95 | 0.42 |
| 12 | 0.00 | −0.45 | 0.34 | −0.95 |
| 13 | −0.01 | 0.46 | 6.10 | 2.43 |
| STOP | 0.00 | 0.00 | −0.01 | 0.00 |
| RSS | 0.02 | 2.49 | 9.91 | 4.11 |

**Table 4.** Desensitized retrofocus lens. Surface sensitivities in waves.

| Desensitized Retrofocus Lens Surface Sensitivities in Waves | | | | |
|---|---|---|---|---|
| Surface | $W_{UA}$ | $W_{UC}$ | $W_{LA}$ | $W_{FT}$ |
| 1 | 0.0 | −0.06 | −0.17 | 0.29 |
| 2 | 0.0 | 0.19 | 1.24 | 0.40 |
| 3 | 0.00 | −0.07 | 0.67 | 0.24 |
| 4 | 0.00 | 0.32 | 1.73 | 1.08 |
| 5 | −0.01 | −0.94 | −0.36 | −0.68 |
| 6 | 0.00 | 0.43 | 1.04 | 0.58 |
| 7 | 0.00 | −0.63 | 0.06 | 0.90 |
| 8 | 0.00 | 0.27 | 0.50 | −0.71 |
| STOP | 0.00 | 0.00 | −0.02 | 0.00 |
| 10 | 0.00 | −0.43 | −2.43 | 0.00 |
| 11 | 0.00 | −0.16 | −0.08 | 0.07 |
| 12 | 0.00 | 0.89 | −0.97 | −0.75 |
| 13 | 0.00 | −0.26 | 1.12 | −0.08 |
| 14 | −0.01 | 0.34 | 4.13 | 1.64 |
| RSS | 0.01 | 1.69 | 5.63 | 2.66 |

### 2.4. Lens Compensation

Often lenses that are mass produced are not adjusted individually to maximize performance because of the time and cost lens adjusting requires. However, if the number of lenses to be made is not too large then it is possible to perform compensation to remove linear astigmatism aberration.

One way to achieve compensation is to roll a lens on its mechanical seat so that one lens surface remains aligned, and the other surface becomes tilted. If $d$ is the lateral displacement of the lens due to rolling, then the tilt $\alpha$ of the surface can be estimated by $\alpha = d/R$, where R is the radius of curvature of the surface resting on the seat. Thus for a radius of curvature of 50 mm and a lens displacement of 0.025 mm the surface tilt becomes about 1.5 arc-minutes.

The sensitivity tables for the above triplet and retrofocus objective lenses show that the last surface of these lenses is much more sensitive to linear astigmatism than to uniform coma. Then by allowing enough diameter clearance for the last lens element of these objectives, it would be possible to compensate for linear astigmatism aberration without significantly introducing uniform coma aberration.

If an objective lens has already been manufactured and suffers from linear astigmatism aberration, then it is possible to add, after the objective, a thin optical wedge to correct for the aberration. The uniform coma and linear astigmatism contributed by a thin optical wedge of apex angle $\alpha$ is given by [19].

$$W_{UC} = -\frac{1}{2}\frac{1-n^2}{n}u^2\alpha y \tag{5}$$

$$W_{LA} = -\frac{1-n^2}{n}u\bar{u}\alpha y \tag{6}$$

where $n$ is the index of refraction of the wedge, $u$ and $\bar{u}$ are the first order marginal and chief rays slopes in image space respectively, and $y$ is the marginal ray height at the wedge. The ratio of linear astigmatism to uniform coma is given by $2\bar{u}/u$. For the triplet lens above the ratio is $2\bar{u}/u = 8.4$, and for the retrofocus lens is also $2\bar{u}/u = 8.4$. The distance between the objective lens and the thin wedge can be used to fine tune the correction because linear astigmatism depends on the marginal ray height at the wedge. However, the wedge apex angle should be minimized to avoid a significant residual of uniform chromatic aberration.

Adding a wedge to correct for linear astigmatism would require measuring the magnitude and orientation of the linear astigmatism and then designing a thin optical wedge. Figure 4 shows a lens arrangement including an optical wedge to correct the linear astigmatism of a perturbed retrofocus lens. The wedge being very weak does not introduce any significant spherical aberration (about 0.1 waves) or uniform chromatic aberration; instead the added wedge will tend to cancel the uniform chromatic aberration from the perturbed lens. A different route to desensitize a retrofocus lens [20] relies on aspherizing some of the lens surfaces.

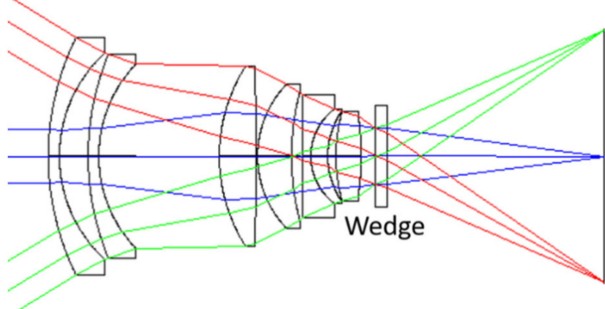

**Figure 4.** Perturbed retrofocus lens and thin optical wedge.

### 2.5. Assembly Trials

Mitigation of manufacturing errors can be achieved by reassembling a lens several times until a better lens is obtained. This method relies on rotating lens elements, interchanging lens elements and lens spacers, and on the statistical nature of the assembling process. Each time an objective lens is reassembled a Monte Carlo assembly trial is performed. Related to this methodology is the concept of selective assembly [21].

As an example consider the telecentric objective lens [22] in Figure 5. This lens has a focal length of 15 mm, a field of view of ±3.5 degrees, and an optical speed of F/1.5. Table 5 provides the sensitivities to aberrations for surface tilts of 0.1°. As can be seen uniform coma dominates followed by linear astigmatism.

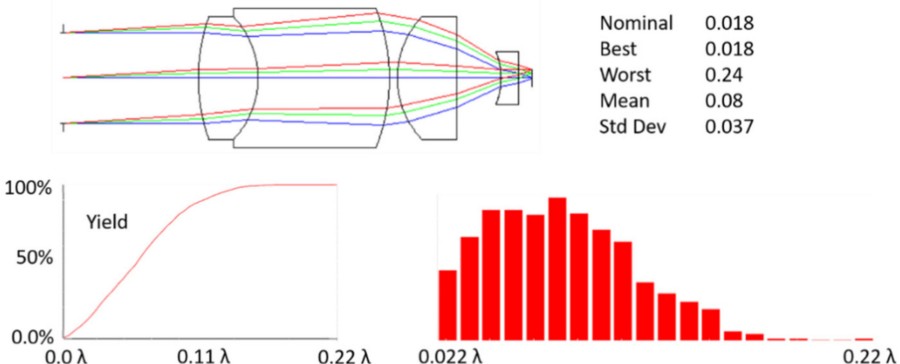

**Figure 5.** Objective lens, yield curve, and histogram for 1000 Monte Carlo trials of rms wavefront error across the field of view.

**Table 5.** Telecentric objective lens. Surface sensitivities in waves.

| Telecentric Objective Lens Surface Sensitivities in Waves | | | | |
|---|---|---|---|---|
| Surface | $W_{UA}$ | $W_{UC}$ | $W_{LA}$ | $W_{FT}$ |
| STOP | 0.00 | 0.00 | 0.00 | 0.00 |
| 2 | 0.00 | −0.11 | 0.20 | 0.12 |
| 3 | 0.00 | −0.65 | 0.37 | 0.19 |
| 4 | −0.01 | 1.18 | −0.73 | −0.40 |
| 5 | 0.00 | −0.28 | 0.21 | 0.22 |
| 6 | 0.00 | 0.54 | −0.33 | −0.30 |
| 7 | 0.00 | −0.19 | 0.23 | 0.21 |
| 8 | 0.00 | 0.06 | 0.02 | −0.07 |
| RSS | 0.01 | 1.50 | 0.96 | 0.64 |

The lens elements tolerances for fabrication were 3 Newton fringes for surface radius of curvature, ±0.05 mm for central lens thickness, and ±5 arc-minutes of lens wedge, except for the second and long lens that was specified with a maximum wedge of ±2.5 arc-minutes. Thirty sets of the lens elements were manufactured.

Based on the tolerances for wedge only, and performing a Monte Carlo analysis of 1000 trials, a histogram vs. wavefront error was produced as shown in Figure 5. The mean lens has a wavefront rms error of 0.08 waves, and the standard deviation is 0.037 waves rms. The yield curve also shown in Figure 5 indicates that 43.3% of the lenses would have a Strehl ratio of 0.08 or larger.

The application of the objective lenses required them to be diffraction limited. The actual objective manufacture involved reassembling the lenses, adjusting lens spacers, rotating lenses, and interchanging lenses. Table 6 provides the performance in terms of Strehl ratio according to the number of objective lenses completed. It was possible to obtain out of the 30 sets, 21 with a Strehl ratio of 0.08 or higher; this is a yield of 70%. Thus assembly trials provide an improved manufacturing yield.

**Table 6.** Objective performance.

| Objective Performance | |
|---|---|
| **Number of Objectives** | **Strehl Ratio** |
| 4 | 0.95 |
| 10 | 0.90 |
| 4 | 0.85 |
| 3 | 0.80 |
| 3 | 0.75 |
| 3 | <0.75 |

## 3. Discussion and Conclusions

This paper discusses the control of linear astigmatism aberration in a perturbed lens system. A set of equations based on real ray tracing is presented to calculate lens sensitivity to aberration. A first way to mitigate linear astigmatism is to desensitize a lens, a second way is compensation, and a third way is by assembling trials. Lens desensitizing involves a trade-off with the nominal performance to achieve a reduced standard deviation, and a low statistical mean. Lens compensation involves rolling a lens that mainly contributes the opposite amount of aberration at the expense of added complexity in the lens barrel and adjustment time. An optical wedge can also be added to correct for linear astigmatism. Assembly trials can provide a yield improvement at the expense of reassembling the lens one or more times. In conclusion, we point out that linear astigmatism aberration prevents a system from being aplanatic and discuss several ways to mitigate this aberration.

**Funding:** This research received no external funding.

**Data Availability Statement:** No new data were created or analyzed in this study. Data sharing is not applicable to this article.

**Acknowledgments:** Many thanks to Richie Youngworth who kindly read and commented on this paper; and to the reviewers who also made valuable suggestions to improve the paper.

**Conflicts of Interest:** The author declares no conflict of interest.

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
