# Peer review of "Control of Linear Astigmatism Aberration in a Perturbed Axially Symmetric Optical System and Tolerancing"

_applsci, doi:10.3390/app11093928_

Round 1
Reviewer 1 Report
I think is a novel and well-written paper, but I would like some parts of it to be clarified.
I don´t understand the script in many words during the manuscript
In Introduction:
the main objective of the study is missing and the new contribution of the paper
In discussion:
Explain the expression "hopefully with a favorable statistical mean"
I think a deeper discussion of the new calculations is necessary, and about what is currently done as well.
I think some final conclusion is necessary
Author Response
Reviewer 1
I thank reviewer 1 for the time and effort put into reviewing the manuscript.
The reviewer concerns are addressed as follows:
- I think is a novel and well-written paper, but I would like some parts of it to be clarified.
I don´t understand the script in many words during the manuscript
In Introduction:
the main objective of the study is missing and the new contribution of the paper
In the introduction and in the discussion sections we point out the goals of the paper and the contributions as follows:
This paper discusses several methods to mitigate linear astigmatism aberration in a system that is nominally axially symmetric but that due to fabrication and assembly errors suffers from this aberration.
Our study focuses on the control of linear astigmatism aberration.
This paper discusses the control of linear astigmatism aberration in a perturbed lens system. A set of equations based on real ray tracing is presented to calculate lens sensitivity to aberration. A first way to mitigate linear astigmatism is to desensitize a lens, a second way is compensation, and a third way is by assembling trials. Lens desensitizing involves a trade-off with the nominal performance to achieve a reduced standard deviation, and a low statistical mean. Lens compensation involves rolling a lens that mainly contributes the opposite amount of aberration at the expense of added complexity in the lens barrel and adjustment time. An optical wedge can also be added to correct for linear astigmatism. Assembly trials can provide a yield improvement at the expense of reassembling the lens one or more times.
In discussion:
Explain the expression "hopefully with a favorable statistical mean"
Thank you for noting this part: we have rewritten it as follows:
Lens desensitizing involves a trade-off with the nominal performance to achieve a reduced standard deviation, and a low statistical mean.
- I think a deeper discussion of the new calculations is necessary, and about what is currently done as well.
We have added the following paragraph:
Keeping in mind that tilting a single surface reduces the system symmetry to plane symmetry, the calculation of uniform coma requires the OPL of two rays in the plane of symmetry. Since lens design software can provide directly the sag of the astigmatic field curves in the plane of symmetry, then the calculation of uniform astigmatism, linear astigmatism, and field tilt is simplified. A discussion of these aberrations is found in reference [4].
I think some final conclusion is necessary
We have added to the conclusion section:
In conclusion, we point out that linear astigmatism aberration prevents a system from being aplanatic and discuss several ways to mitigate this aberration.

Reviewer 2 Report
This manuscript discusses different methods to mitigate linear astigmatism aberration in systems axially symmetric, but due to fabrication and assembly erros the aberrations are induced.
The manuscript show a large ammount of typo errors:
Title:
Line 2 per-turbed->perturbed
Line 3 toleranc-ing->tolerancing
Abstrac:
Line 13 sys-tems ->system
Introduction
Line 17 sharpim-aging-> sharp imaging
Line 21 Ap-lanatism->aplanatism
....
Lines 26,28,29,38,39,48,52,71,78,79,93,99... and so many more.
The manuscript must be carefully reviewed by the author.
In section 2 (Materials and methods) some physical discussions I think are required, explaining for example the influence of the tilted amount election in the aberrations. Which are the effects if different tilt amount is induced at different surfaces?
It is also important to explain the relations of coefficients W_uc, W_ua, W_la...with the aberrations coefficients associated to Zernike polynomials usually employed in software for optical desing.
In this sense the author do not inform in the manuscript about the software used in order to obtain the results. It is not possible to repeat exactly the numerical experiences carried out by the author with the information reported, so this issue must be revised adding supplementary materials.
Comparison with other authors results will be interesting, for example the Cooke triplet lens analyzed in section 2.1 have been recently discused in reference:
Design of optical systems that maximize asbuilt performance using
tolerance/compensator-informed optimization
BRIAN J. BAUMAN AND MICHAEL D. SCHNEIDER
Vol. 26, No. 11 | 28 May 2018 | OPTICS EXPRESS 13819
When in retrofocus lens (or any other system) is included a thin optical wedge to correct astigmatism aberration, could be interesting to deeply discuss not only uniform chromatic aberration but also the spherical aberration induced to the total system.
There are references that should be included:
Ellis I. Betensky "Aberration correction and desensitization of an inverse-triplet object lens", Proc. SPIE 3482, International Optical Design Conference 1998, (21 September 1998); https://doi.org/10.1117/12.322011
John S. Petersen "Optical proximity strategies for desensitizing lens aberrations", Proc. SPIE 4404, Lithography for Semiconductor Manufacturing II, (26 April 2001); https://doi.org/10.1117/12.425214
Author Response
Reviewer 2
I thank reviewer 2 for the time and effort put into reviewing the manuscript.
The reviewer concerns are addressed as follows:
- The manuscript show a large ammount of typo errors:
Title:
Line 2 per-turbed->perturbed
Line 3 toleranc-ing->tolerancing
Abstrac:
Line 13 sys-tems ->system
Introduction
Line 17 sharpim-aging-> sharp imaging
Line 21 Ap-lanatism->aplanatism
....
Lines 26,28,29,38,39,48,52,71,78,79,93,99... and so many more.
The manuscript must be carefully reviewed by the author.
The typos have been corrected as the manuscript has been reviewed.
- In section 2 (Materials and methods) some physical discussions I think are required, explaining for example the influence of the tilted amount election in the aberrations. Which are the effects if different tilt amount is induced at different surfaces?
We have added the following paragraph:
The choice of 0.1° surface tilt is because this amount of surface tilt often separates what is easy to manufacture from what is difficult to manufacture. The sensitivity analysis could be done with a larger surface tilt and the results would be similar because uniform coma, linear astigmatism, and field tilt are proportional to surface tilt.
- It is also important to explain the relations of coefficients W_uc, W_ua, W_la...with the aberrations coefficients associated to Zernike polynomials usually employed in software for optical desing.
We point out that:
Depending on how the data is processed Zernike polynomials or other type of polynomials can be used. However, here for clarity we chose a simple aberration representation.
- In this sense the author do not inform in the manuscript about the software used in order to obtain the results. It is not possible to repeat exactly the numerical experiences carried out by the author with the information reported, so this issue must be revised adding supplementary materials.
We have added that:
The analyses discussed below were done with Zemax and OpticStudio lens design software.
- Comparison with other authors results will be interesting, for example the Cooke triplet lens analyzed in section 2.1 have been recently discused in reference:
Design of optical systems that maximize asbuilt performance using
tolerance/compensator-informed optimization
BRIAN J. BAUMAN AND MICHAEL D. SCHNEIDER
Vol. 26, No. 11 | 28 May 2018 | OPTICS EXPRESS 13819
We have added the statement:
Previously the desensitized form of a Cooke triplet in Fig. 1 at right has been reported [17]; here we note that the main problem is linear astigmatism.
- When in retrofocus lens (or any other system) is included a thin optical wedge to correct astigmatism aberration, could be interesting to deeply discuss not only uniform chromatic aberration but also the spherical aberration induced to the total system.
We now state that:
The wedge being very weak does not introduce any significant spherical aberration (about 0.1 waves) or uniform chromatic aberration; instead the added wedge will tend to cancel the uniform chromatic aberration from the perturbed lens. A different route to desensitize a retrofocus lens [20] relies on aspherizing some of the lens surfaces.
There are references that should be included:
Ellis I. Betensky "Aberration correction and desensitization of an inverse-triplet object lens", Proc. SPIE 3482, International Optical Design Conference 1998, (21 September 1998); https://doi.org/10.1117/12.322011
John S. Petersen "Optical proximity strategies for desensitizing lens aberrations", Proc. SPIE 4404, Lithography for Semiconductor Manufacturing II, (26 April 2001); https://doi.org/10.1117/12.425214
These references have been added.
